# Development of a Deep-Learning-Based Artificial Intelligence Tool for Differential Diagnosis between Dry and Neovascular Age-Related Macular Degeneration

**DOI:** 10.3390/diagnostics10050261

**Published:** 2020-04-28

**Authors:** Tae-Young Heo, Kyoung Min Kim, Hyun Kyu Min, Sun Mi Gu, Jae Hyun Kim, Jaesuk Yun, Jung Kee Min

**Affiliations:** 1Department of Information and Statistics, Chungbuk National University, Chungdae-ro 1, Seowon-gu, Cheongju-si, Chungbuk 28644, Korea; 2College of Pharmacy and Medical Research Center, Chungbuk National University, 194-31 Osongsaengmyeong 1-ro, Osong-eup, Heungdeok-gu, Cheongju-si, Chungbuk 28160, Korea; 3Department of Ophthalmology, Ulsan University Hospital, College of Medicine, University of Ulsan, 877, Bangeojinsunhwando-ro, Dong-gu, Ulsan 44033, Korea

**Keywords:** age-related macular degeneration, class activation map, convolutional neural network, cross-validation, retina

## Abstract

The use of deep-learning-based artificial intelligence (AI) is emerging in ophthalmology, with AI-mediated differential diagnosis of neovascular age-related macular degeneration (AMD) and dry AMD a promising methodology for precise treatment strategies and prognosis. Here, we developed deep learning algorithms and predicted diseases using 399 images of fundus. Based on feature extraction and classification with fully connected layers, we applied the Visual Geometry Group with 16 layers (VGG16) model of convolutional neural networks to classify new images. Image-data augmentation in our model was performed using Keras ImageDataGenerator, and the leave-one-out procedure was used for model cross-validation. The prediction and validation results obtained using the AI AMD diagnosis model showed relevant performance and suitability as well as better diagnostic accuracy than manual review by first-year residents. These results suggest the efficacy of this tool for early differential diagnosis of AMD in situations involving shortages of ophthalmology specialists and other medical devices.

## 1. Introduction

Deep-learning-based artificial intelligence (AI) tools have been adopted by medical experts for disease diagnosis and detection. Furthermore, AI-based diagnosis can be used to augment human analysis in pathology and radiology [1]. AI-based tools have been developed for cancer diagnosis using pathology slides, preliminary radiology reports, chest X-rays, and detection of cardiac dysfunction using electrocardiograms [2,3,4]. Image extraction and analytical algorithms in AI diagnosis are drawing the attention of medical specialists. For example, use of deep learning tools to analyze photographs of lesions represents a potential methodology for diagnosing several retinal diseases in ophthalmology. Therefore, deep learning AI tools might be useful to ophthalmologists for predicting and treating diabetic retinopathy, age-related macular degeneration (AMD), floaters, and retinitis pigmentosa. Recently, the Food and Drug Administration approved IDx-DR for the detection of diabetic retinopathy as an AI-based diagnostic system [5]. In diabetic retinopathy, blood vessels become blocked and irregular in diameter [6], which induces fluid leakage and hemorrhaging associated with vision damage. Additionally, angiogenesis is considered to be a pathogenic process in diabetic retinopathy [7]. These pathologies represent potential photographic sources for the development of AI diagnosis tools.

AMD is a degenerative eye disease and the leading cause of irreversible vision loss in the elderly [8]. It is a complex, multifactorial disease, and the pathogenesis is not fully understood. [9]. Choroidal neovascularization (CNV), vascular leakage, and hemorrhaging are the hallmarks of neovascular AMD (nAMD) [10]. Detection of AMD in its early stages is important for a good prognosis, and the differential diagnosis between dry AMD (dAMD) and nAMD is also critical for appropriate treatment and reduction of disease severity [11]. However, a shortage of ophthalmologists and medical devices for diagnosis represents a potential challenge for the timely detection of diseases.

Ophthalmologists can diagnose AMD through eye examinations, such as fundus photography, optical coherence tomography (OCT), fluorescein angiography (FA), and indocyanine green angiography (ICGA) [12], with multimodal imaging also potentially necessary for accurate AMD diagnosis and treatment. However, for diagnostic screening purposes, it is difficult to access all of the various imaging equipment. Fundus photography has the limitation of providing only two-dimensional retinal information. However, it is an inexpensive and relatively simple device-based diagnostic tool that is easy to operate. Additionally, images can be saved and used at a later time by different clinicians and researchers. Furthermore, this method results in higher patient compliance due to the short test times and non-invasiveness of the method.

Fundus photographs record the appearance of patient retinas, allowing the clinician to detect retinal changes and review the findings with a colleague [13]. AMD-related leakage of fluid and blood can be observed by fundus photography, which is also capable of detecting drusen, mottled appearance, and hemorrhagic detachment. Therefore, fundus photography might be useful for diagnosing AMD in routine eye examinations.

In this study, we explored the viability of fundus photography for the development of a deep-learning-based AI diagnostic tool and demonstrated the performance of the proposed AI tool for differentially diagnosing AMD (control vs. dAMD vs. nAMD). Additionally, we compared the diagnostic accuracy of the AI tool with that of ophthalmology residents for AMD.

## 2. Materials and Methods

### 2.1. Ethical Approval

This study was reviewed, and the protocol approved by the Institutional Human Experimentation Committee Review Board of Ulsan University Hospital, Ulsan, Republic of Korea (UUH 2019-12-006, 31 December 2019). The study was conducted in accordance with the ethical standards set forth in the 1964 Declaration of Helsinki.

### 2.2. Subjects

To select patient groups (nAMD and dAMD), the medical records of patients aged >50 years who had been diagnosed with nAMD or dAMD between March 1, 2015, and July 31, 2019, at the Department of Ophthalmology of Ulsan University Hospital, Ulsan, Republic of Korea, were retrospectively reviewed. All subjects (399 eyes of 378 patients) underwent a complete ophthalmic examination that included the best-corrected visual acuity assessment, non-contact tonometry (CT-1P; Topcon Corporation, Tokyo, Japan), and swept-source OCT (DRI OCT-1 Atlantis; Topcon Corporation, Tokyo, Japan). CNV or polypoidal vascular lesions were detected via FA and ICGA (Heidelberg Retina Angiograph Spectralis; Heidelberg Engineering, Heidelberg, Germany). Patients who had had previous retinal surgeries, such as epiretinal membrane, macular hole, vitreous hemorrhage, and rhegmatogenous retinal detachment (RRD), were excluded. Subjects were also excluded if they had pre-existing ocular diseases (such as glaucoma, uveitis, diabetic retinopathy, and retinal vascular disease, which are known to affect retinal pathophysiology), severe media opacity, or high myopia (axial length ≥ 26.5 mm). To select a normal control, the medical records of patients who had been diagnosed with and surgically treated for various retinal diseases (macular hole, epiretinal membrane, or RRD) were also reviewed. Normal control was defined based on the absence of lesions, including drusen, according to fundus photography and OCT in the unaffected eyes.

### 2.3. Imaging Equipment

We used two fundus photography systems (TRC-NW8, Topcon Corporation, Tokyo, Japan, and Daytona, Optos, Inc., Marlborough, MA, USA). The TRC-NW8 retinal camera provides high-quality 16.2-megapixel images, with a 45° field of central macular view. Daytona provides ultra-widefield fundus digital images at 200° of the retina in a single pass. All retinal images were reviewed by a retinal specialist (JKM) to ensure that the photographs were of sufficiently high quality to adequately visualize the retina.

### 2.4. Convolutional Neural Network (CNN) Modeling

Convolutional neural network (CNN) techniques have recently shown noticeable advances in various fields, including computer vision and image analysis. We used this method to classify macular degeneration in macular images. We used a modified Visual Geometry Group with 16 layers (VGG16) model [14] (the winner of the ILSVRC-2014 competition) as a deep learning model for classification. VGG16 has a very simple architecture that uses only 3 × 3 convolutional layers and 2 × 2 pooling layers (Figure 1). We loaded the VGG16 model with image datasets from ImageNet (http://www.image-net.org/) and trained the convolutional layers and fully connected layers with macular images. The macular image dataset was divided into two sets, with 30% of the images in each group placed into the test set, and the remaining images used for the training set [15]. Training was performed using multiple iterations with a learning rate of 0.000001 and Nadam optimization.

Class activation map (CAM) visualization was performed to identify areas displaying the greatest effect of macular degeneration. CAM extracts feature maps of the final convolutional layer (Conv5_3) of the model trained using macular images and computes the weights of the feature maps to represent the heatmap in the image.

### 2.5. Preprocessing

Each original image had a resolution of 913 × 837 pixels with a 24-bit RGB channel. We first identified the appropriate coordinates for cropping the images to ensure that they were centered around the center of the macula. The coordinates were then adjusted to eliminate unnecessary information, such as the black margin area. All images were cropped based on fixed and adjusted coordinates, with the cropped images having a resolution of 500 × 500 pixels. All of the cropped images were then resized to 244 × 244 pixels as input images for the deep learning model. Preprocessed images were generated using various methods with Keras ImageDataGenerator (https://keras.io/) during training.

### 2.6. Cross-Validation of Artificial Intelligence (AI)-Based Diagnosis

Cross-validation is a useful technique for evaluating the performance of deep learning models. In cross-validation, the dataset is randomly divided into training set and test sets, with the training set used to build a model, and the test set used to assess the performance of the model by measuring accuracy.

In k-fold cross-validation, the dataset is divided randomly into k subsets of equal size, with one used as a test set and the others for training. The cross-validation is performed k times to allow for the use of all subsets exactly once as a test set. Model performance is determined according to the average of model evaluation scores calculated across k test subsets. Here, we evaluated the performance of the proposed CNN model using 5-fold cross-validation, with performance determined according to the average accuracy of five cross-validations for each class comparison.

### 2.7. Comparative Analysis of Accuracy Values of the AI Diagnosis Tool and Residents in Ophthalmology

To compare the performance between AI diagnosis and that of clinical reviewers, two residents in our hospital evaluated the fundus images used to develop the tool. Reviewer 1 was a first-year resident and reviewer 2 a fourth-year resident in ophthalmology. For 3-class classification, control, dAMD, and nAMD fundus photos were randomly displayed on a computer screen for 20 s, and the two reviewers interpreted fundus findings as Normal, dAMD, or nAMD. For 2-class classification, comparisons were divided into three groups (Normal vs. dAMD, Normal vs. nAMD, and dAMD vs. nAMD) according to the same time constraints. The fundus photos were also randomly displayed on the screen for 20 s. Two reviewers read retina findings as Normal or dAMD in the Normal–dAMD group, as Normal or nAMD in the Normal–nAMD group, and as dAMD or nAMD in the dAMD–nAMD group. Accuracy values for each reviewer were calculated and presented accordingly.

## 3. Results

### 3.1. Fundus Image Collection

Eyes (*n* = 142) from 126 patients were diagnosed with nAMD, and fundus images were collected. Fundus examination of eyes with nAMD can include one or more features, such as subretinal and/or intraretinal fluid, subretinal hemorrhage, and retinal pigment epithelial detachment and intraretinal exudate in the macular area (Figure 2). Based on the category of AMD in age-related eye diseases [16], drusen types corresponding to categories 2 and 3 were defined as dAMD (132 eyes from 127 patients) through fundus photography and OCT (Figure 3a,b). Furthermore, images of 125 eyes from 125 patients were collected as controls (Figure 3c,d).

### 3.2. Data Augmentation

We performed several iterative learning steps using Nadam optimization, and loss values of the model were recorded for each iteration. The model with the lowest loss value recorded during training was adopted and used. The images were processed using Keras ImageDataGenerator, with the center of the macula located in the center of the image. Image generation was performed by randomly moving in the up, down, left, and right directions, flipping, and applying image rotation and zooming according to a previous report [17] (Figure 4).

### 3.3. Validation of the Deep-Learning-Based Diagnostic Tool

CAM visualization showed that the convolutional neural network (CNN) successfully identified areas of degeneration (Figure 5). These represent the most important areas in each image in the trained CNN when classified as AMD. In the case of dAMD, the drusen, which resembled a yellow dot characteristic of dAMD symptoms, was correctly identified. For nAMD, areas involving degeneration and bleeding were identified, with pathological changes, such as elevation, observed in the center of the macula. Accordingly, we were able to identify macular morphological changes characteristic of nAMD (Figure 5).

We achieved 90.86% accuracy with preprocessing for three-class classification. Table 1 shows a comparison of accuracy between a preprocessed (w-Pre) model and a non-preprocessed (w/o-Pre) model. The results indicated that the w-Pre model performed better in terms of accuracy, except for the comparison of the control with dAMD, and that preprocessing of fundus images improved classification. Table 2 and Table 3 show the detailed results for each fold. Therefore, we used the modified VGG-16 model trained with preprocessed data.

Table 2 shows the accuracies and the average accuracies obtained for each fold of cross-validation, with the accuracies for two-class classification higher than those for three-class classification. Table 4 shows measurements of sensitivity, specificity, positive predictive value, and negative predictive value, revealing a similar outcome of two-class classification outperforming three-class classification. Additionally, we performed the same validation using ultra-widefield images (Table 5), resulting in similar results. Furthermore, while using ultra-widefield images, the tool showed an accuracy of 0.7584 in normal vs. dAMD, 0.9099 in normal vs. nAMD, and 0.7601 in dAMD vs. nAMD for the two-class classification. In the three-class classification of ultra-widefield images, the accuracy was 0.7321. We evaluated the performance of medical reviewers and compared this with outcomes using AI diagnosis (Table 6). The results showed that the AI tool outperformed both a first- and fourth-year resident in accurately differentiating between AMD types and a control for both three-class and two-class classifications.

## 4. Discussion

Recent advances in deep learning techniques have increased the focus of medical specialists on potential application of AI-based diagnostic tools. Given the image-extraction features of deep learning algorithms, these techniques are potentially suitable for analyzing photographs from eye examinations. AMD is a leading cause of vision loss, and early detection is important for a good prognosis [18]. Furthermore, differential diagnosis between dAMD and nAMD is critical for suitable treatment [19]. However, based on the shortage of ophthalmologists and medical devices, early diagnosis of dAMD and nAMD is challenging. In this study, we developed a deep-learning-based diagnostic tool to detect and differentiate between dAMD and nAMD using fundus photographs. To the best of our knowledge, this represents the first development and application of an AI tool for differential diagnosis of AMD type.

Five-fold cross-validation revealed that our AI model showed high accuracy (>0.9) for both three-class and two-class classification and comparable and superior accuracy to diagnoses by medical reviewers (fourth- and first-year residents). Additionally, differential diagnosis using ultra-widefield images from AMD patients revealed overall accuracies lower than those obtained using conventional fundus images. Unlike conventional fundus images, the ultra-widefield images were not formal photos and included unnecessary information (eyelids, light bleeds, different pixels, etc.), making manual preprocessing of the images necessary. These results suggested that ultra-widefield images were not appropriate for use with deep learning tools.

Our AI tool detected features of AMD, such as drusen, bleeding, and elevation of the center of the macula. Specifically, bleeding and degeneration of the centers of maculae are markers used for nAMD diagnosis [20]. Multimodal imaging (e.g., OCT) is generally necessary for accurate AMD diagnosis and prognostic prediction. The low reliability of diagnostic imaging equipment can result in a poor diagnosis and prognosis, especially in low-income countries. Therefore, we developed an AI tool for AMD diagnosis that uses only conventional fundus photographs and demonstrated the efficacy of the tool for differential diagnosis between dAMD and nAMD. Our findings support this AI tool as a cost-effective methodology that addresses possible shortages of eye specialists and medical devices required for accurate AMD diagnosis.

## Figures and Tables

**Figure 1 diagnostics-10-00261-f001:**
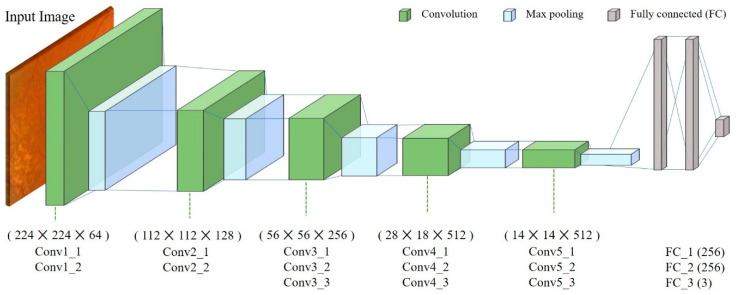
The proposed convolutional neural network (CNN) architecture (a modified Visual Geometry Group with 16 layers (VGG16) model). The CNN with the modified VGG16 model used 3 × 3 convolutional layers and 2 × 2 pooling layers. Convolutional layers and fully connected layers were trained with macular images.

**Figure 2 diagnostics-10-00261-f002:**
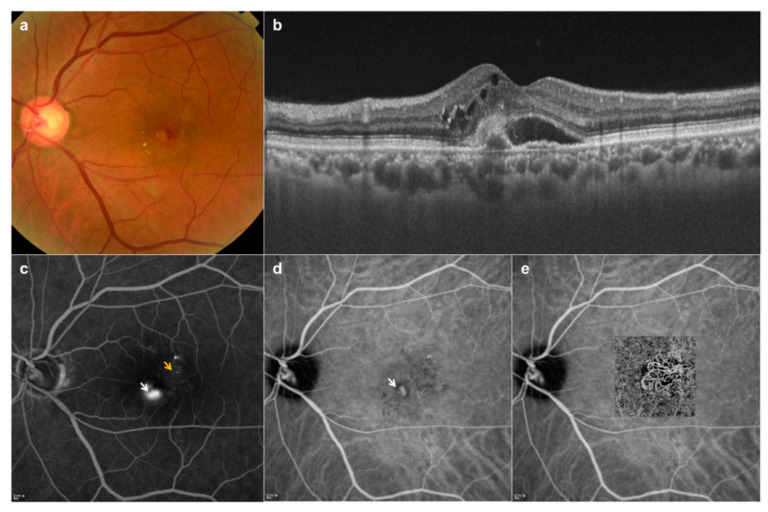
Multimodal images of neovascular age-related macular degeneration in a 61-year-old man. (**a**) Fundus photography shows subretinal fluid, exudation, and hemorrhage; (**b**) Optical coherence tomography (OCT) B-scan revealed non-uniform hyper-reflective formations above the retinal pigment epithelium and the presence of intraretinal cysts and subretinal fluid; (**c**) Fluorescein angiography (FA) demonstrates aspects of a well-defined (white arrow) and an irregular (yellow arrow) hyper-fluorescent lesion; (**d**) Indocyanine green angiography (ICGA) shows staining of the type 2 choroidal neovascularization (CNV) (white arrow); (**e**) An OCT angiography image (with the neovascular network) overlaid on the ICGA image.

**Figure 3 diagnostics-10-00261-f003:**
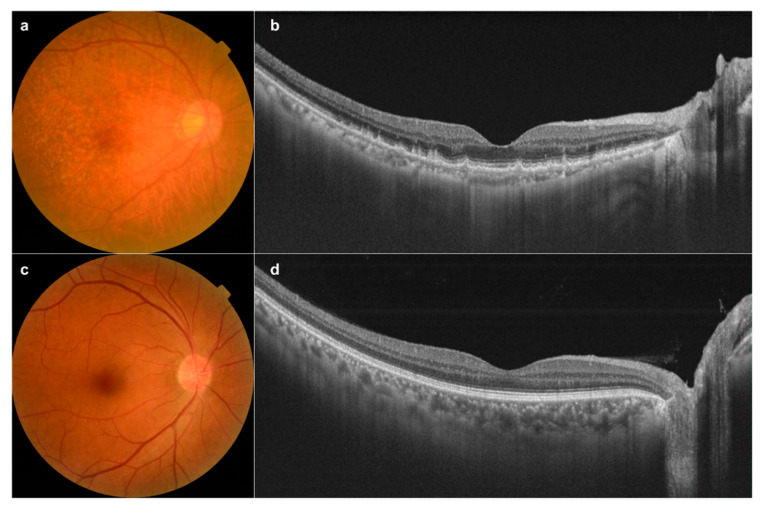
Fundus photography and optical coherence tomography of dry age-related macular degeneration (dAMD) and control retinas. (**a**) Numerous soft, yellow drusen in the right eye of a 78-year-old woman; (**b**) The corresponding OCT image shows multiple deposits accumulating under the retinal pigment epithelium. (**c**) Normal control fundus photography in the right eye of a 66-year-old man. (**d**) The corresponding OCT image of the control.

**Figure 4 diagnostics-10-00261-f004:**
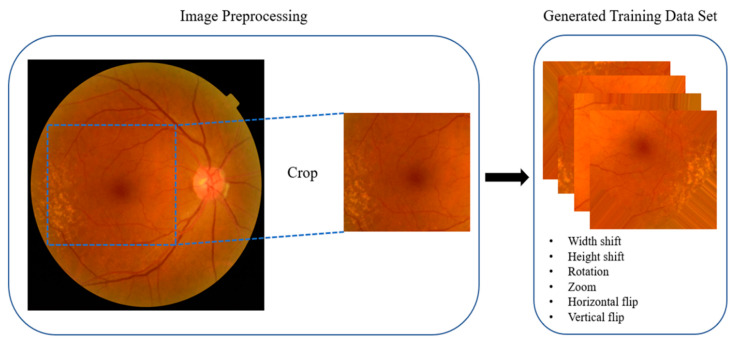
Image preprocessing. Eye images were preprocessed by Keras ImageDataGenerator. Original images were cropped and resized to 244 × 244 pixels. The training dataset images were generated using various methods, including width shift, height shift, rotation, zoom, horizontal flip, and vertical flip.

**Figure 5 diagnostics-10-00261-f005:**
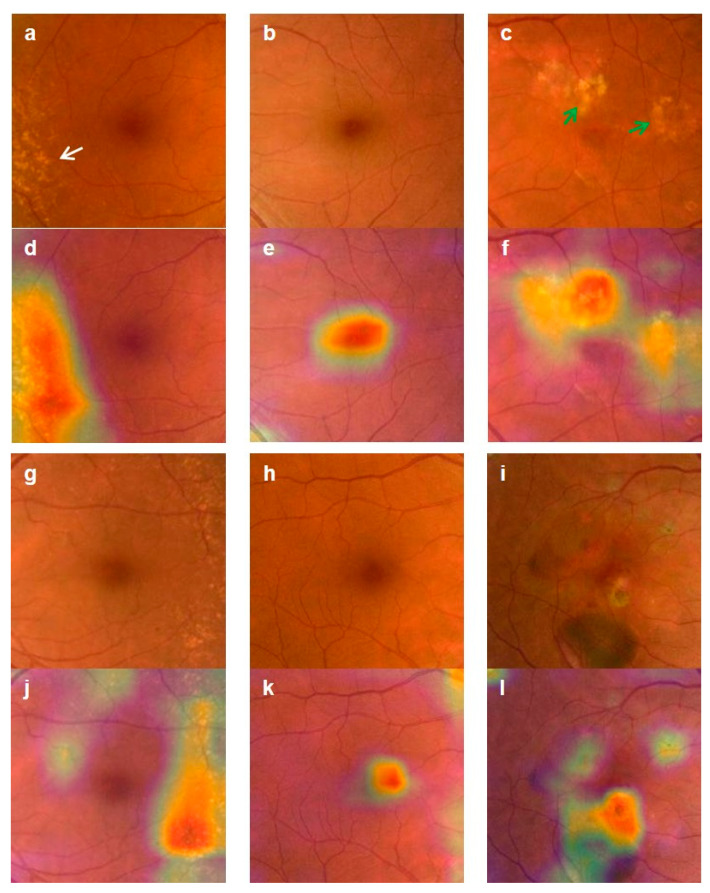
Examples of class activation map (CAM) visualization. CAM visualization of normal, dry age-related macular degeneration (dAMD), and neovascular age-related macular degeneration (nAMD) retinas. CAM extracts the feature map of the last convolution layer (Conv5_3) and shows a heatmap within the image describing the calculated weight of the feature map. (**a**) dAMD fundus images show drusen (arrow), and (**d**) heatmap images show drusen identified by the artificial intelligence (AI) tool; (**b**) Normal fundus images have no drusen, and (**e**) heatmap images of normal controls show that the AI tool identified the contour of fovea according to the absence of drusen; (**c**) nAMD fundus images show bleeding and degenerated areas (green arrows), and (**f**) heatmap images show identified drusen and other features of degeneration and bleeding; (**g**–**i**) Representative images of dAMD, a normal control, and nAMD, respectively; (**l**) Heatmap images of nAMD show that the AI tool identified pathological changes in the macula, such as elevation of the center; (**j**) There was no heatmap at the center of dAMD; however, the AI tool detected drusen instead; (**k**) Heatmap image showing AI identification of the center of the macula in a control, with no degenerated area.

**Table 1 diagnostics-10-00261-t001:** Comparison of outcomes according to preprocessing.

Average Accuracy	3-Class	2-Class		
	Control–dAMD–nAMD	Control–dAMD	Control–nAMD	dAMD–nAMD
w-Pre	0.9086	0.9192	0.9813	0.9132
w/o-Pre	0.8559	0.9264	0.9808	0.9063

Data represent calculated accuracy values.

**Table 2 diagnostics-10-00261-t002:** Results obtained using five-fold cross-validation with preprocessing.

Folds	3-Class	2-Class		
	Normal–dAMD–nAMD	Normal–dAMD	Normal–nAMD	dAMD–nAMD
Fold 1	0.9756	0.8846	1.0000	0.9231
Fold 2	0.8864	1.0000	1.0000	0.8929
Fold 3	0.9535	0.9259	1.0000	1.0000
Fold 4	0.9318	0.9286	1.0000	0.9643
Fold 5	0.7955	0.8571	0.9063	0.7857
Average	0.9086	0.9192	0.9813	0.9132

Data represent calculated accuracy values.

**Table 3 diagnostics-10-00261-t003:** Results obtained using five-fold cross-validation without preprocessing.

Folds	3-Class	2-Class		
	Normal–dAMD–nAMD	Normal–dAMD	Normal–nAMD	dAMD–nAMD
Fold 1	0.8049	0.8846	0.9667	0.9615
Fold 2	0.8409	0.9286	1.0000	0.8571
Fold 3	0.8837	0.9259	1.0000	0.9626
Fold 4	0.8864	0.9643	1.0000	0.8214
Fold 5	0.8636	0.9286	0.9375	0.9286
Average	0.8559	0.9264	0.9808	0.9063

Data represent calculated accuracy values.

**Table 4 diagnostics-10-00261-t004:** Average classification results for each model.

Model		Accuracy	Sensitivity	Specificity	PPV	NPV
3-class		0.9086	0.9046	1.0000	1.0000	0.9349
	Control–dAMD–nAMD		0.8605	0.9394	0.8303	0.9500
			0.9571	0.9329	0.8750	0.9786
2-class	Control–dAMD	0.9192	0.9252	0.9167	0.8788	0.9492
	Control–nAMD	0.9813	0.9684	1.0000	1.0000	0.9625
	dAMD–nAMD	0.9132	0.8795	0.9448	0.9318	0.8992

Data represent calculated accuracy values. PPV, positive predictive value; NPV, negative predictive value.

**Table 5 diagnostics-10-00261-t005:** Results obtained using five-fold cross-validation and ultra-widefield images.

Folds	3-Class	2-Class		
	Normal–dAMD–nAMD	Normal–dAMD	Normal–nAMD	dAMD–nAMD
Fold 1	0.7885	0.6071	0.8636	0.8636
Fold 2	0.7885	0.7143	0.8182	0.7727
Fold 3	0.6481	0.8571	0.9130	0.6957
Fold 4	0.7500	0.7857	0.9545	0.7727
Fold 5	0.6852	0.8276	1.0000	0.6957
Average	0.7321	0.7584	0.9099	0.7601

Data represent calculated accuracy values.

**Table 6 diagnostics-10-00261-t006:** Comparison of differential diagnosis of AMD type between first- and fourth-year residents.

Folds	3-Class		2-Class					
	Normal–dAMD–nAMD		Normal–dAMD		Normal–nAMD		dAMD–nAMD	
	Reviewer 1	Reviewer 2	Reviewer 1	Reviewer 2	Reviewer 1	Reviewer 2	Reviewer 1	Reviewer 2
Fold 1	0.7317	0.9024	0.9615	0.9231	0.9667	1.0000	0.6923	0.9231
Fold 2	0.7045	0.9091	0.9643	0.8929	0.9375	0.9062	0.8519	0.9259
Fold 3	0.6977	0.8140	0.8519	0.9259	0.9375	0.8750	0.8148	0.9630
Fold 4	0.7955	0.7273	0.9643	0.9286	0.9062	0.9688	0.7500	0.9286
Fold 5	0.7143	0.8000	0.7500	0.9643	0.8750	0.9062	0.7143	0.7143
Average	0.7287	0.8306	0.8984	0.9270	0.9246	0.9312	0.7647	0.8910

Reviewers 1 and 2 represent a first- and fourth-year resident, respectively. Data represent calculated accuracy values.

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
