# Peer review of "Development of a Deep-Learning-Based Artificial Intelligence Tool for Differential Diagnosis between Dry and Neovascular Age-Related Macular Degeneration"

_diagnostics, 2020, doi:10.3390/diagnostics10050261_

Round 1
Reviewer 1 Report
In this study, Heo et al. investigated the accuracy of a deep learning algorithm in distinguishing normal retinae from retinae with either dry age-related macular degeneration (AMD) or neovascular AMD. I congratulate the authors for their efforts. I hope that my comments below can help improve the already well-written manuscript.
1. Introduction, line 49-51 "However, choroidal neovascular (CNV) is responsible for more than 80 % of vision loss in AMD patients." This line can be confusing, at least from the perspective of a clinician. I recommend omitting that specific sentence.
2. The authors describe an excellent clinical protocol for diagnosis. For example, it is described that ICGA is performed, which would be important in diagnosing polypoidal choroidal vasculopathy (PCV), which the authors also state in the manuscript. But it seems unclear whether cases with PCV were allowed in the analysis or not. Such cases have slightly different appearance on fundus photographies than neovascular AMD, and therefore it is important to know whether or not PCVs were allowed when understanding the results. Please elaborate.
3. From a clinical and practical perspective, I recommend that the authors present more data than just accuracy: True positive, true negative, false positive, false negative, sensitivity, and specificity. Such measures are crucial when evaluating diagnostic performance.
Reviewer 2 Report
This article reports on the development of a deep learning tool for diagnosis of wet and dry AMD.
Images used were highly processed (cropping, registration). A very small number of original images were used, which were then apparently used as the basis for producing many derived images by applying minor transformations of the originals.
Results as reported are unremarkable, roughly matching that of a 1st year ophthalmology resident given 20 seconds to examine each image.
QUESTION: did the human observers view the original images, or the cropped and processed images? This is not clear from the paper.
The paper claims that fundus photography is "relatively simple" and "easy to operate for learners", while at the same time pointing out that considerable pre-processing is required to remove common imaging artifacts in order to make the images suitable for their system.
No information is given on the test/retest performance of the human graders.
In this reviewer's opinion, the conclusion that this tool is suitable for diagnostic imaging in low-income countries is not warranted by the data presented. I agree with their statement that "For accurate diagnosis and prediction of prognosis of AMD, multimodal imaging (e.g. OCT) is necessary".
It is not at all clear why it is useful to have a screening tool which is primarily aimed at distinguishing between wet and dry AMD. Much more relevant is a tool that accurately distinguishes normal vs diseased eyes - or, even better, identifies those AT RISK for developing either form of AMD.
This is a reasonably competent exercise in applying deep learning tools to fundus photos, but it does not provide convincing numbers, and the justification and relevance are very weak.
Major flaws include: low number of training images, heavy (manual) pre-processing of actual fundus photos. and unsurprising results.
Minor flaws: the Introduction contains many claims that are probably correct, but which should be supported by specific citations.
That said, the paper does adequately explain what was done, and the results achieved.
Round 2
Reviewer 2 Report
The authors have been mostly responsive to my concerns. While I still have some minor quibbles, I think it is acceptable in its current form.